# Carbon Nanodots as Electron Transport Materials in Organic Light Emitting Diodes and Solar Cells

**DOI:** 10.3390/nano13010169

**Published:** 2022-12-30

**Authors:** Zoi Georgiopoulou, Apostolis Verykios, Kalliopi Ladomenou, Katerina Maskanaki, Georgios Chatzigiannakis, Konstantina-Kalliopi Armadorou, Leonidas C. Palilis, Alexander Chroneos, Evangelos K. Evangelou, Spiros Gardelis, Abd. Rashid bin Mohd Yusoff, Athanassios G. Coutsolelos, Konstantinos Aidinis, Maria Vasilopoulou, Anastasia Soultati

**Affiliations:** 1Institute of Nanoscience and Nanotechnology (INN), National Center for Scientific Research Demokritos, Agia Paraskevi, 15341 Athens, Greece; 2Solid State Physics Section, Physics Department, National and Kapodistrian University of Athens, Panepistimioupolis, Zografos, 15784 Athens, Greece; 3Department of Chemistry, International Hellenic University, 65404 Kavala, Greece; 4Department of Physics, University of Ioannina, 45110 Ioannina, Greece; 5Department of Physics, University of Patras, Rio, 26504 Patra, Greece; 6Department of Electrical and Computer Engineering, University of Thessaly, 38221 Volos, Greece; 7Department of Materials, Imperial College, London SW7 2AZ, UK; 8Department of Chemical Engineering, Pohang University of Science and Technology (POSTECH), Pohang 37673, Gyeongbuk, Republic of Korea; 9Laboratory of Bioinorganic Chemistry, Department of Chemistry, University of Crete, Voutes Campus, Heraklion, 71003 Crete, Greece; 10Department of Electrical and Computer Engineering, Ajman University, Ajman P.O. Box 346, United Arab Emirates; 11Center of Medical and Bio-allied Health Sciences Research, Ajman P.O. Box 388, United Arab Emirates

**Keywords:** organic light emitting diodes, organic solar cells, carbon nanodots, surface functionalization, electron transport materials

## Abstract

Charge injection and transport interlayers play a crucial role in many classes of optoelectronics, including organic and perovskite ones. Here, we demonstrate the beneficial role of carbon nanodots, both pristine and nitrogen-functionalized, as electron transport materials in organic light emitting diodes (OLEDs) and organic solar cells (OSCs). Pristine (referred to as C-dots) and nitrogen-functionalized (referred to as NC-dots) carbon dots are systematically studied regarding their properties by using cyclic voltammetry, Fourier-transform infrared (FTIR) and UV–Vis absorption spectroscopy in order to reveal their energetic alignment and possible interaction with the organic semiconductor’s emissive layer. Atomic force microscopy unravels the ultra-thin nature of the interlayers. They are next applied as interlayers between an Al metal cathode and a conventional green-yellow copolymer—in particular, (poly[(9,9-dioctylfluorenyl-2,7-diyl)-alt-co-(1,4-benzo-{2,1′,3}-thiadiazole)], F8BT)—used as an emissive layer in fluorescent OLEDs. Electrical measurements indicate that both the C-dot- and NC-dot-based OLED devices present significant improvements in their current and luminescent characteristics, mainly due to a decrease in electron injection barrier. Both C-dots and NC-dots are also used as cathode interfacial layers in OSCs with an inverted architecture. An increase of nearly 10% in power conversion efficiency (PCE) for the devices using the C-dots and NC-dots compared to the reference one is achieved. The application of low-cost solution-processed materials in OLEDs and OSCs may contribute to their wide implementation in large-area applications.

## 1. Introduction

In 1987, Tang and Van Slyke introduced the first electroluminescent device based on organic semiconductors, the so-called organic light emitting diode (OLED) [1]. After three decades, they are considered a mature technology and have already achieved successful market entry. However, OLEDs still attract research interest because of their potential as a promising technology for solid state lighting and flat panel displays [2], considering their low-cost fabrication, flexibility and large viewing angle. The balance of charge injection and the transport of holes and electrons through the organic semiconductor plays a vital role regarding the device’s (OLED and OSC) efficiency [3]. One of the key issues for the efficient design of organic optoelectronic devices is the understanding of the energy-level alignment at the metal contact/organic semiconductor interfaces. Proper matching of the electrode Fermi level to the energy level of the charge transport states of the organic semiconductor is necessary to obtain efficient and balanced charge injection in these organic electronic devices [4]. This is why, in highly efficient devices, anode and cathode interfacial layers are necessary to enhance the charge exchange between metal contacts and organic molecules. 

A typical structure of an OLED and/or OSC is based on a multilayer architecture [5]. The emissive and/or photoactive layer and the various interlayers (hole injection layers (HILs), hole transport layers (HTLs), electron blocking layers (EBLs), hole blocking layers (HBLs), electron transport layers (ETLs) and electron injection layers (EILs)) are sandwiched between the two electrodes (anode and cathode). It is well known that in organic electroluminescent devices, the mobility of holes is larger than that of electrons and the hole injection barrier is lower than the electron injection barrier [6]. Such devices have extra holes in the emission region, and electrons are more or less consumed before reaching the emissive layer, which decreases the device’s efficiency. An effective approach is to modify the cathode, aiming to reduce the energy barrier and increase the charge transport between the metal cathode and the emissive and/or photoactive layer. This can be accomplished by inserting suitable interfacial layers (ETL, HBL, EIL) [7,8]. A large number of organic and inorganic materials have been applied as ETLs in OLEDs and OSCs to lower the electron injection barrier and achieve charge carrier balance. They include transition metal oxides with intrinsic n-type conductivity, such as zinc oxide (ZnO) [9,10,11,12] and tin dioxide (SnO_2_) [13,14], and mostly organic molecules bearing suitable functional groups, such as perylene [15,16] and porphyrin [17,18,19] compounds, polymeric oxadiazoles, metal chelates, azole-based materials and triazine [20,21,22,23,24,25]. However, despite their great success, many of these materials require vacuum deposition methods, which are incompatible with large-area devices for low-cost applications. The development of solution-processable materials of potentially low cost and with facile synthesis and deposition methods still represents a topic of intense research interest in OLEDs and OSCs.

Carbon nanodots are advantageous in terms of the abundant and sustainable precursor materials and facile surface functionalization to promote their beneficial properties. The facile and low-cost preparation/modification of these materials motivated us to further explore their use as ETLs in high-performance green emitting OLEDs and inverted OSCs. We synthesized pristine and nitrogen-functionalized carbon dots (described hereafter as C-dots and NC-dots, respectively) and unraveled their effects on the OLED and OSC performance, which paves the way for the further exploration of not only organic but also perovskite optoelectronics. 

## 2. Materials and Methods

### 2.1. OLED Device Fabrication Procedure

The first and most fundamental step before performing the layers’ deposition was the proper cleaning of the glass substrates. We utilized coated glass substrates of indium-tin oxide (ITO) purchased from Sigma-Aldrich (Athens, Greece), with sheet resistance of 15–25 Ω/sq, which served as the transparent anode electrode. ITO-coated glasses were placed into an ultrasonic cleaner, where they remained for ten minutes in each bath: first in deionized water, then in acetone and finally in isopropyl alcohol (IPA). Moreover, all samples were dried with N_2_ gas before each bath. Then, the ITO substrates were subjected to UV–ozone treatment for 20 min. Next, we performed the deposition of hole injection layers (HTLs) using the commercially available solution PEDOT:PSS (poly(3,4 - ethylenedioxythiophene)−poly(styrenesulfonate)) with 1.3 wt % dispersion in H_2_O, from Sigma-Aldrich. At first, it was passed through a 0.45 μm pore diameter polyvinylidene fluoride (PVDF) filter and then was spin-coated on ITO-coated glass substrates at 6000 rpm for 40 s, forming a 40 nm thick layer. The substrates were annealed at 110 °C for at least 30 min on a hotplate. A green-yellow copolymer, named F8BT (poly[(9,9-dioctylfluorenyl-2,7-diyl)-alt-co-(1,4-benzo-{2,1′,3}-thiadiazole)), purchased from the American Dyes Source, Quebec, Canada, (ADS 233 YE), used as the emissive layer. F8BT was filtered through a 0.22 μm pore diameter PTFE filter and was deposited on top of the PEDOT:PSS. The deposition was carried out by spin-coating at 1200 rpm for 40 s from a 10 mg mL^−1^ solution in chloroform, forming a 80 nm thick layer. The substrates were annealed at 85 °C for 10 min on a hotplate. Moreover, carbon dots were employed as EILs. In particular, 50 μL of carbon dots (C-dots) and nitrogen-doped carbon dots (NC-dots) was spin-coated at 2000 rpm for 40 s from a 0.5 mg mL^−1^ solution in methanol atop the emissive layer. Finally, the OLED devices were completed with the deposition of a 150 nm aluminum layer through thermal evaporation in order to serve as the cathode electrode. The active surface of each diode was set at 12.56 mm^2^. 

### 2.2. OSC Device Fabrication Procedure 

Glass/ITO substrates were cleaned as already mentioned and used as the transparent cathode electrode for the fabrication of the inverted OSCs. Tin oxide purchased from Alfa Aesar was deposited from a water solution (15% colloidal dispersion of tin(vi) oxide in deionized water was diluted in 1:6.5 water) with spin-coating at 3000 rpm for 30 s. Then, the samples were thermally annealed at 150 °C for 30 min. PTB7-Th:PCBM (purchased from Ossila, Sheffield, UK), at 10 mg:15 mg in 1 mL 1,2-dichlorobenzene with the addition of 3% per volume 1,8 diiodoctane (DIO), was spin-coated at 1000 rpm for 90 s on the SnO_2_ ETL to serve as the photoactive layer. The fabrication procedure was completed with the deposition of molybdenum oxide (MoO_x_) HTL by a hot-wire CVD method [26] and Al anode electrode by thermal evaporation. 

### 2.3. Characterization Methods

Current density–voltage–luminance (Ι−V−L) measurements were performed using a Keithley 2601A Power Supply Source-Meter (Vector Technologies, Athens, Greece) in voltage mode, with constant increment steps. The instrument was used both for slow (delay time of 1 sec before each measurement point) and fast (delay time of <1 msec) measurements. EL analysis was performed using a calibrated photodiode (BPW34 Si PIN photodiode) (Vector Technologies, Athens, Greece) connected to a Keithley 6500 DMM (Vector Technologies, Athens, Greece). The same instruments were used for the current and luminance–time (I-t and L-t) measurements. In this case, the current measurements through the diodes were acquired at rates of up to 1K sample/sec, while the luminance values were obtained at faster rates, reaching values of up to 1M sample/sec. All measurements were performed at room temperature in a dark shielded probe station to minimize light and EM interference. For the OSCs’ electrical characterization in dark and illuminated conditions, a Keithley 2400 source meter unit and a Xe lamp with an AM 1.5G filter were used. TEM measurements were performed using a Philips CM 20 transmission electron microscope (Thermo fisher scientific, United States). UV–Vis absorbance and transmittance spectra were captured by a Perkin Elmer Lambda 40 UV–Vis spectrometer (Vamvakas-Scientific Equipment, Athens, Greece). XRD patterns were recorded using a Smart Lab Rigaku diffractometer (Japan) with Cu Kα radiation. Cyclic voltammetry measurements were performed using a VersaSTAT 4 potentiometer (Megalab SA, Athens, Greece). Fourier-transform infrared (FT-IR) spectroscopy was performed using a Bruker Tensor 27 spectrometer (Interactive, Athens, Greece), with a DTGS detector. The surface morphology of devices was recorded with an NT-MDT AFM system (LaborScience SA, Athens, Greece) in tapping operation mode. Ellipsometry measurements were carried out with a J.A Woolam Inc. M2000F rotating compensator ellipsometer that operated within the 250–1000 nm range, running the WVASE32 software, at an angle of incidence of 75.14°. Steady-state PL measurements were performed with a commercial platform (ARKEO—Cicci Research). In particular, the substrate was illuminated with a diode-pumped solid-state Nd:YVO4 + KTP laser (peak wavelength 532 nm ± 1 nm, optical power 1 mW on a circular spot of 2 mm of diameter 31 mW cm^−2^) at an inclination of 45°. The fluorescence on the opposite side of the substrate was focused on a bundle of fibers (10 mm in diameter) with an aspheric lens close to the substrate to maximize the PL. The bundle sent the signal to a CCD-based spectrometer. The integration time and the number of averages was maintained to better compare the results. Time-resolved PL (TRPL) spectra were measured with an FS5 spectrofluorometer from Edinburgh Instruments (Livingston, UK). A 478.4 nm laser was used as an excitation source. All measurements were performed in environmental conditions at room temperature.

## 3. Results and Discussion

### 3.1. Characterization of Carbon Dots and F8BT/Carbon Dot Interfaces

In this work, we studied carbon and nitrogen-functionalized carbon nanodots that were synthesized by a bottom-up approach from inexpensive, organic precursors; their design was based on sodium carboxylate groups and carboxylic acid with amino-terminated groups, respectively. The surface terminal groups of the synthesized carbon dots (C-dots) and nitrogen-doped carbon dots (NC-dots) are illustrated in Figure 1a. They were both processed from solutions in orthogonal to organic semiconductor solvents, such as methanol and dimethylformamide (DMF).

The morphology and structure of these nanodots and carbon dots were investigated by transmission electron microscopy (TEM). Figure 2a shows a TEM image of the C-dots, suggesting the formation of uniform spherical particles with an average size of around 3 nm. The crystallinity of the nanodots was studied by X-ray diffraction (XRD) measurements. Figure 2b,c show the XRD patterns of thin film deposited from C-dot and NC-dot solutions, respectively. In the case of C-dots, the only peak that appeared was assigned to the silicon substrate; therefore, the sample was amorphous. However, the diffraction pattern of NC-dots consisted of more intense peaks that revealed the formation of a more crystalline structure in these dots. As represented in Figure 2c, the XRD pattern of the NC-dots included peaks at 2*θ* = 20.9°, 28.7°, 39.0° and 61.6°. The corresponding Miller indices according to the relevant literature are also shown in Figure 2c,d [27]. Notably, XRD revealed that the NC-dot sample was more crystalline compared to the pristine C-dots. The lattice d-spacing was found to be 0.31 nm. This could be beneficial for electron transport when these materials are used as cathode interfacial layers in organic optoelectronic devices. Figure 2d depicts the UV–Vis absorption spectra and the derived Tauc plot for the estimation of the energy bandgap (E_G_) values of these materials. The extracted values are 2.6 eV for C-dots and 2.3 eV for the NC-dots. 

Amino groups have been considered beneficial for use in EILs as they usually induce the formation of negative interfacial dipoles at the cathode interface [28]. As a result, they can reduce the electron injection barrier at the respective contact. Therefore, it is of significant importance to estimate the energy levels of the synthesized materials and obtain valuable information about the energetic alignment at the cathode interface. As cyclic voltammetry is considered one of the most powerful tools to investigate the reduction and oxidation processes of molecular species, we employed this method to estimate the highest occupied molecular orbital (HOMO) and lowest unoccupied molecular orbital (LUMO) energy levels of these materials. Figure 2e,f show the cyclic voltammograms of C-dot and NC-dot films, respectively, deposited on indium-tin oxide (ITO)/glass substrates. 

Oxidation and reduction processes are represented, allowing the HOMO and LUMO calculation, respectively, using the empirical formulas [29,30]: E_HOMO_ = −(E_ox,onset_ + 4.4) eV (1)
E_LUMO_ = −(E_red,onset_ + 4.4) eV (2)
where E_ox,onset_ and E_red,onset_ are the oxidation and reduction potential onset, respectively, defined as the position where the current starts to differ from the baseline. From the cyclic voltammograms the extracted E_ox,onset_ value is +1.2 V and +1.8 V for C-dots and NC-dots, respectively, which results in HOMO levels of −5.6 eV for C-dots and −6.2 eV for NC-dots. The LUMO levels of C-dots and NC-dots are −3.4 eV and −3.5eV, respectively, determined by the reduction potential onset (−1 V for C-dots and −0.9 V for NC-dots).

The deposited films were further characterized by Fourier-transform infrared spectroscopy (FT-IR) in order to recognize any possible changes in the F8BT film after the deposition of C-dots and NC-dots (Figure 3). In the C-dot spectrum, the broad band observed in the range of 3500-3100 cm^−1^ corresponds to the stretching vibration of the O–H bond, while the weak band at 2930 cm^−1^ corresponds to C–H stretching. The two bands at 1560 cm^−1^ and 1395 cm^−1^ correspond to the asymmetric and symmetric stretching of the carboxylate group. In the case of the NC-dot spectrum, the stretching modes of N–H, O–H and C–H are detected in the broad band ranging from 3600 cm^−1^ to 2800 cm^−1^. The other bands detected at 1700 cm^−1^, 1650 cm^−1^ and 1560 cm^−1^ refer to the C=C stretching mode, the C=O stretching mode and the N-H bending mode, respectively. 

As far as the spectrum of the F8BT emitter is concerned, the bands at 2930 cm^−1^ and 2850 cm^−1^ correspond to the stretching mode of the C–H bond, while the band at 1460 cm^−1^ is due to the C=C stretching mode of an aromatic group [31,32]. Furthermore, the stretching mode between the phenyl rings is detected at 1250 cm^−1^, the deformation of aliphatic chain bonds at 1100 cm^−1^ and the C–H rocking mode at 812 cm^−1^. In the case of the F8BT films with C-dots deposited on top, a slight increase in the intensity and width of the band at 1100 cm^−1^ is observed, with no other obvious changes. When NC-dots are deposited on top of a F8BT film [32], a similar change is apparent in the 1100 cm^−1^ region, where the band corresponding to aliphatic chain deformation almost disappears. This is a sign of possible interaction between the aliphatic chains of the F8BT molecule and the NC-dot material deposited on top. However, the bands corresponding to the stretching vibrations of the same aliphatic chains do not change, which is an indication of the limited extent of this interaction.

We next investigated the optoelectronic and morphological properties of F8BT before and after coverage with thin carbon dot interlayers (processed from 0.5 mg mL^−1^ in methanol solutions). In Figure 4a, the UV−Vis absorption spectra of F8BT without and with carbon dots spin-coated are presented. All these spectra exhibit absorption peaks centered at wavelengths of 372 nm and 450 nm, corresponding to F8BT [33]. The characteristic carbon dot absorption peaks were not detected in layers deposited on top of the F8BT, which is an indication that they form very thin interlayers on top of F8BT. 

Moreover, the emission characteristics of the same samples were investigated by measuring the steady-state photoluminescence (PL) spectra (Figure 4b). While the spectra of F8BT and F8BT/C-dots are identical, a shoulder observed at approximately 580 nm for the F8BT/NC-dot sample can be attributed to an interaction between the aliphatic chains of the F8BT molecule and the NC-dot interlayer, as already discussed in the FT-IR measurements. In addition, no changes in the time-resolved photoluminescence (TRPL) spectra (Appendix A) of pristine and nanodot-coated F8BT films are observed, exhibiting similar average lifetimes in all cases. However, the extinction coefficients and refractive indices (RI) of F8BT pristine and coated with carbon dots present some changes (Figure 4c,d). In particular, a change in the RI may be an indication that the insertion of these dots can change the light outcoupling efficiency of the fabricated OLEDs. Such change, however, is expected to be rather marginal since the difference in RI is very small at the emission wavelength range of F8BT (centered on 530 nm). Notably, the carbon dot interlayers induce a small increase in the surface roughness of F8BT, as indicated by the similarity of the atomic force microscopy (AFM) surface topography of these samples (Figure 5). A small increase in surface roughness or the formation of small nanostructures may be beneficial for the device performance as they enlarge the area, where electron injection towards the emissive layer occurs [34,35,36].

### 3.2. OLED Measurements

To demonstrate the practical utility of these solution-processed carbon nanodots, we next applied them as EIL/ETLs in fluorescent OLEDs based on a green-yellow fluorescent polymer, well known as F8BT. The accurate structure of the OLEDs is represented in Figure 6a. Specifically, the devices consisted of indium-tin oxide (ITO) coated glass substrates as the transparent anode electrode, poly(3,4-ethylenedioxythiophene) polystyrene sulfonate (PEDOT:PSS) as the hole transport layer, F8BT as the emissive layer, C-dots or NC-dots as the electron transport layers and Al contact as the cathode electrode. A reference device using a commonly used solution-processed EIL, namely cesium carbonate (Cs_2_CO_3_), was also fabricated. The corresponding energy level diagram of the materials embedded into the OLED structure, as well as the chemical structure of F8BT, is illustrated in Figure 6b. Note that the work functions of ITO, PEDOT:PSS and Al, along with the energy levels of F8BT, were taken from the literature [37,38]. The HOMO and LUMO levels of both carbon nanodots were estimated by cyclic voltammetry and optical measurements, as already mentioned.

Figure 6c shows the electroluminescence (EL) spectra of the three different types of devices. It becomes obvious that the devices with C-dots and NC-dots do not affect the spectral dependence of the emission, which originates only from F8BT. Figure 6d illustrates a plot of the current density versus the applied voltage and luminance versus the applied voltage of the OLEDs. Devices with C-dots and, especially, those using NC-dots have a rapid increase in the current density at the low-voltage regime—an indication of the reduced electron injection barrier in these devices compared to the reference one. Furthermore, from the plot of luminance versus the applied voltage of these OLEDs, the device turn-on voltage (V_ON_, the voltage where the luminance becomes equal to 1 cd m^−2^) can be estimated. For the reference device, the V_ON_ is calculated to be around 8 V, whereas it is decreased to nearly 6 V and further to 4 V for the devices using C-dots and NC-dots, respectively, indicating the crucial role of the carbon dots in the device performance. Figure 6e,f present the luminous efficiency (LE) and external quantum efficiency (EQE) of the prepared OLEDs, where a clear enhancement in the device performance is observed when nanodots, especially NC-dots, are incorporated in the device. In particular, the NC-dot-based OLED exhibits a ~1.5 fold higher LE and EQE compared with the reference device. A small enhancement in hydrophobicity induced by the coverage of F8BT with the carbon nanodots (Appendix A) is also observed, which could be beneficial in the device’s stability.

Furthermore, to unambiguously demonstrate the beneficial effect of these carbon dots for OLEDs, mainly due to a reduction in the electron injection barrier, we fabricated electron-only devices and measured the carrier obtained by blocking one type of carrier (holes) through the absence of the hole injection layer (PEDOT:PSS) at the anode side of the device. Figure 7 shows the current density taken in electron-only devices using either Cs_2_CO_3_ or the newly developed carbon nanodots at one side of the device. By considering that the bulk-limited currents are similar to these devices, the obtained current densities are governed by electrode-limited processes (electron barriers are responsible for conduction) [16,39]. As shown in Figure 7, the current density versus voltage (J-V) curve of the reference diode (shown in a semilog plot) is always lower compared to the current densities of the carbon dots using diodes, a clear indication that the electron injection barrier is reduced upon the application of carbon nanodots, especially those with the amino-functionalized surface, as cathode interlayers in these devices. 

### 3.3. OSC Measurements

In order to strengthen our approach, we also employed the nanodots as cathode interlayers in inverted OSCs based on a PTB7-Th (poly[4,8-bis(5-(2-ethylhexyl)thiophen-2-yl)benzo[1,2-b;4,5-b’]dithiophene-2,6-diyl-alt-(4-(2-ethylhexyl)-3-fluorothieno[3,4-b]thiophene-)-2-carboxylate-2-6-diyl)]) donor and PCBM ([6,6]-phenyl-C61-butyric acid methyl ester) acceptor blended photoactive layer. Figure 8a illustrates the device structure of the fabricated inverted OSCs, while Figure 8b presents the chemical structure of the donor and acceptor materials that constituted the photoactive layer. Carbon nanodots spin-coated from a methanol solution with a concentration of 0.5 mg mL^−1^ on tin oxide (SnO_2_) were used as ETLs, forming an ultra-thin film to improve the electron transport between the cathode and the photoactive layer. Note that molybdenum oxide (MoO_x_), used as a hole transport layer, and a reference device without the nanodots were also considered for comparison. Figure 8c shows the transmittance spectra of the pristine and nanodot-coated SnO_2_. It is observed that the nanodot interlayers slightly affect the transmittance and absorption (Appendix A) properties of SnO_2_ in the visible region, suggesting no prevention of sunlight reaching the photoactive layer, which could lead to a current density reduction. Accordingly, no changes in energy bandgap values (E_G_ = 3.65 eV) are observed for SnO_2_ films coated with or without carbon dots (Appendix A). 

Figure 8d represents the energy level diagram of the materials used in the inverted OSCs. Note that, except the energy levels of the carbon nanodots, the work functions and energy levels of the SnO_2_, PTB7-Th and PCBM were taken from previous work [40,41]. As in the case of OLEDs, a reduction in the electron extraction barrier at the cathode/PCBM interface is observed, verifying the beneficial role of the nanodots coated atop the SnO_2_ film as cathode interlayers in the device performance. The current density–voltage (J-V) characteristic curves of the nanodot-based and reference OSCs are depicted in Figure 8e. A ~13% improvement in the device efficiency is observed for the OSCs with the nanodot interlayers, revealing the universality of the carbon dots used as ETLs in organic optoelectronic devices. In particular, the NC-dot-based device exhibited a short-circuit current density (J_SC_) of 11.14 mA cm^−2^, open-circuit voltage (V_OC_) of 0.78 V and fill factor (FF) of 0.52, resulting in a higher PCE of 4.52% compared with the reference cell, showing a PCE of 3.94% (J_SC_ of 10.66 mA cm^−2^, V_OC_ of 0.77 V and FF of 0.48). The device with the C-dots showed also higher electrical parameters (J_SC_ of 11.08 mA cm^−2^, V_OC_ of 0.78 V and FF of 0.51) and thus efficiency (PCE of 4.41%) with respect to the reference device.

## 4. Conclusions

We have demonstrated here the beneficial role of carbon nanodots with different surface functionalization in the performance of solution-processed green OLEDs and inverted OSCs. These materials were investigated for their optoelectronic properties and possible interaction with the organic semiconductors. Our experimental data indicate that they induce significant reductions in the electron injection/extraction barrier, therefore resulting in the fabrication of OLEDs and OSCs with significantly reduced turn-on voltages and increased short-circuit current density, respectively. Our results demonstrate the beneficial effects of abundant, carbon-based materials in organic optoelectronics for large-area applications.

## Figures and Tables

**Figure 1 nanomaterials-13-00169-f001:**
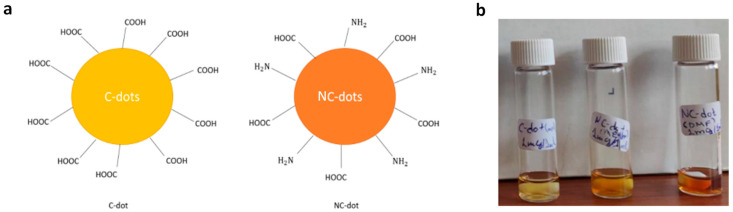
(**a**) Illustration of the chemical structures (surface functionalization) of C-dots and NC-dots. (**b**) Photographs demonstrating the solution processability of carbon nanodots. From left to the right: a methanol solution of C-dots, a methanol solution of NC-dots and a DMF solution of NC-dots.

**Figure 2 nanomaterials-13-00169-f002:**
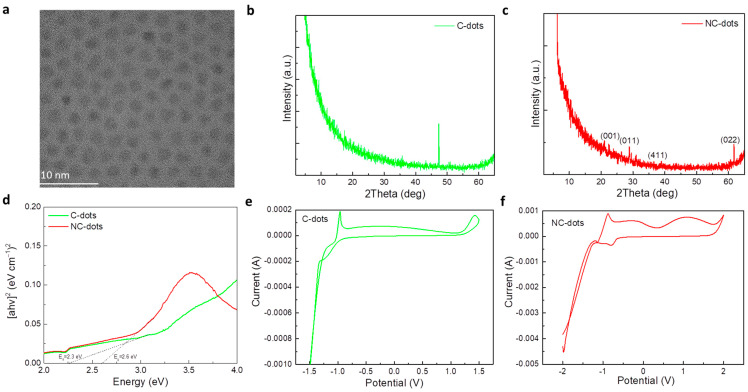
(**a**) Transmission electron microscopy (TEM) images of carbon dots. XRD patterns of (**b**) C-dot and (**c**) NC-dot films drop cast on silicon substrates from concentrated (10 mg mL^−1^) methanol solutions. (**d**) Tauc plots of C-dot and NC-dot films drop cast onto quartz substrates from concentrated (10 mg mL^−1^) methanol solutions. Cyclic voltammetry of (**e**) C-dot and (**f**) NC-dot films coated on indium-tin oxide (ITO)/glass substrates at a scan rate of 0.1 V s^−1^ in a 0.1 M LiClO_4_ aqueous electrolyte solution.

**Figure 3 nanomaterials-13-00169-f003:**
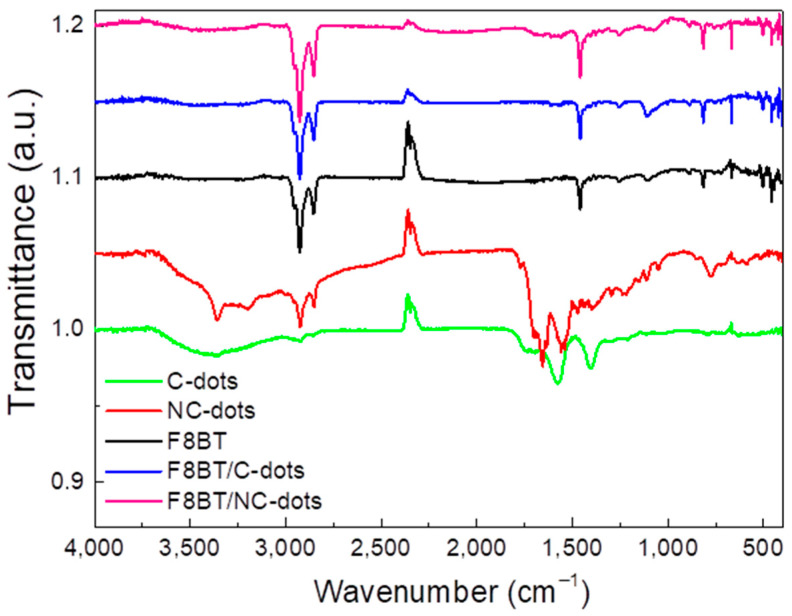
FTIR spectra of pristine carbon dots and F8BT films and of F8BT/C-dot and F8BT/NC-dot interfaces.

**Figure 4 nanomaterials-13-00169-f004:**
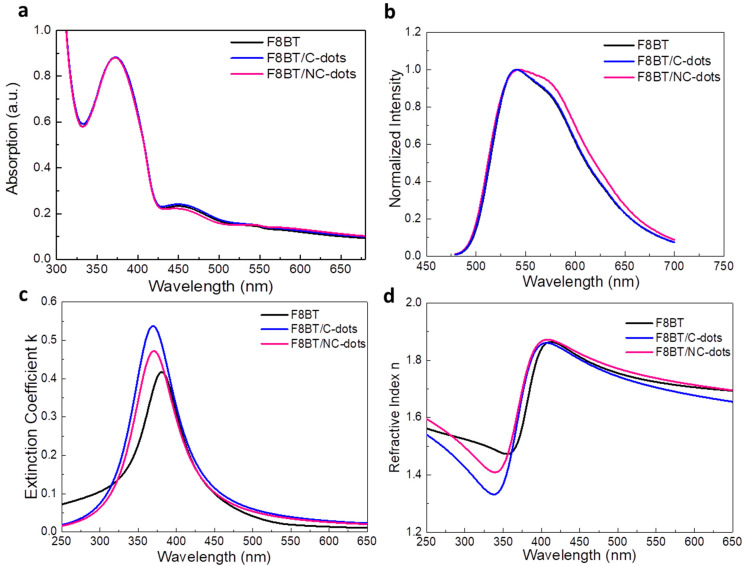
(**a**) UV–Vis absorption and (**b**) steady-state PL spectra of F8BT pristine and coated with C-dots and NC-dots. (**c**) Extinction coefficient and (**d**) refractive index measurements of F8BT and F8BT/C-dot and F8BT/NC-dot interfaces.

**Figure 5 nanomaterials-13-00169-f005:**
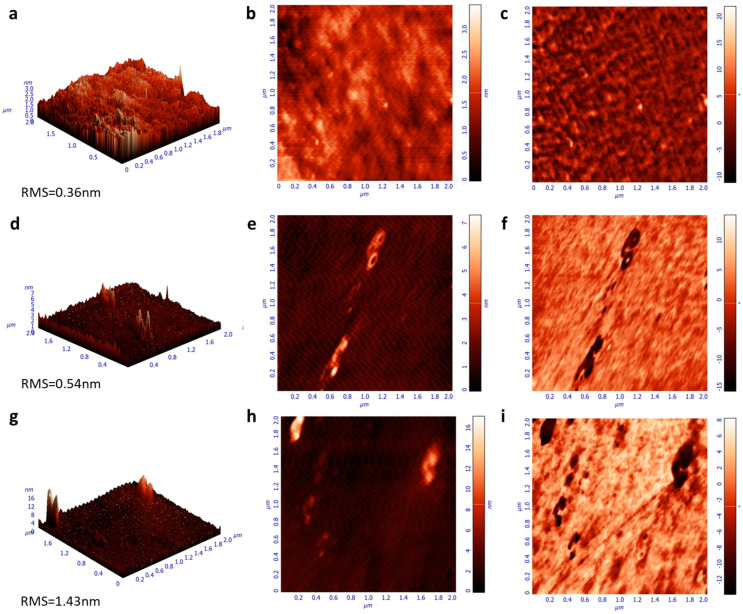
Atomic force microscopy (AFM) surface topographies (height, 3D-left and 2D-middle, phase-right) of (**a**–**c**) F8BT, (**d**–**f**) F8BT/C-dot and (**g**–**i**) F8BT/NC-dot films.

**Figure 6 nanomaterials-13-00169-f006:**
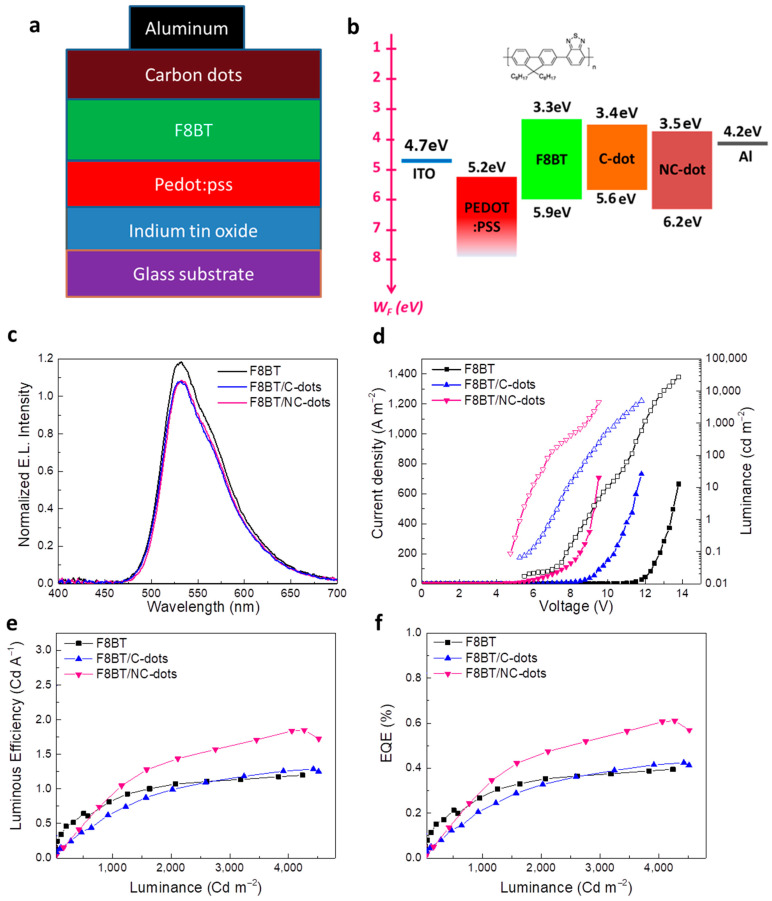
(**a**) The OLED architecture and (**b**) the corresponding energy level diagram. The chemical structure of the F8BT is also presented. (**c**) Electroluminescence (EL) spectra of the different devices at a voltage of 5 V. (**d**) Current density (solid symbols)–voltage and luminance (open symbols)–voltage characteristics of the three types of OLEDs. (**e**) Luminous efficiency (LE) and (**f**) external quantum efficiency (EQE) of the same devices.

**Figure 7 nanomaterials-13-00169-f007:**
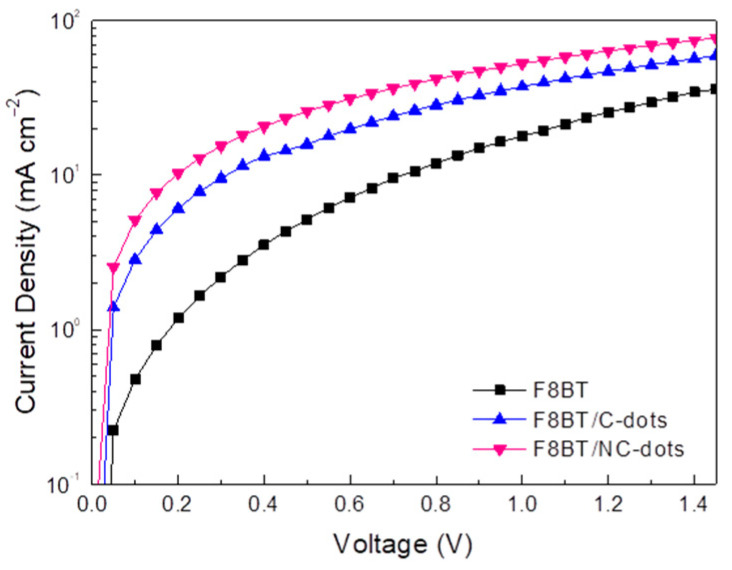
Electron-only current densities of diodes with the structure ITO/F8BT/Al without and with C-dots and NC-dots at the cathode side of the device.

**Figure 8 nanomaterials-13-00169-f008:**
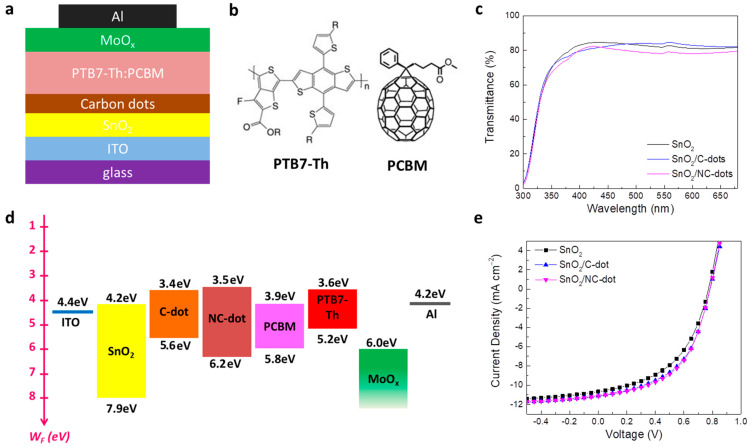
(**a**) Schematic representation of the fabricated inverted OSCs. (**b**) Chemical structure of the donor and acceptor materials used in the photoactive blended layer. (**c**) Transmittance spectra of the SnO_2_ film coated with or without the C-dots and NC-dots. (**d**) Energy level diagram of the different layers used for the fabrication of the inverted OSCs. (**e**) Current density–voltage (J-V) characteristic curves measured under AM 1.5G illumination.

## Data Availability

The data that support the findings of this study can become available by the corresponding authors upon reasonable request.

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
