# Peer review of "Carbon Nanodots as Electron Transport Materials in Organic Light Emitting Diodes and Solar Cells"

_nanomaterials, 2022, doi:10.3390/nano13010169_

Round 1
Reviewer 1 Report
The manuscript by Georgiopoulou et al. describes experiments that aim at an improvement of organic light emitting diodes (OLEDs) and organic photovoltaic cells by inclusion of carbon nanoparticle (C-dots) based interlayers. Two differently functionalized nanoparticles have been investigated.
The manuscript has several technically weak points that, apart from a number of grammar issues, need to be clarified or corrected. These include the following:
1. Some abbreviations are defined or explained only several pages after they have been first used, such as F8BT (first occurrence in line 40 vs 213/214 with a brief and lines 320-322 with a longer definition).
2. The achieved 'increase in power conversion efficiency (PCE) over 10%' (lines 43/44) sounds impressive but is a relative one, from an external quantum efficiency of only ≈0.4% to ≈0.55% according to figure 6f.
3. The statement that 'mobility of holes is larger than that of electrons and hole injection barrier is lower than electron injection barrier' (lines 66/67) is not generally valid but applies only to certain types of OLEDs.
4. The interpretation of all parts of figure 2 seems highly debatable:
a) I cannot see in Figure 2a 'C-dots revealing a diameter of about 2-3nm' as stated in line 105. The bright field image is very poor and shows the edge of some rather compact specimen where it is unclear what the slight mottled intensity variations are. Please show a clearer image of dispersed particles on some thin carbon foil support.
b,c) The XRD spectra show only 1 peak each, at completely different angles, and the conclusion in line 108 is incorrect as long as it cannot be ruled out that these peak come from the underlying silicon substrates. Please show better XRD data and label all peaks properly with Miller indices and elements.
d) How the dotted curves in the UV-vis spectra can yield highly precise band-gap measurements is totally unclear. Please provide useful Tauc plots instead.
e, f) The interpretation of the voltammograms seems suspect:
i. If I follow the instruction in lines 126-127 to define the values of the onset of the oxidation potential by looking at baseline deviations, I would get values around -1eV or +1.2eV for e) and -1eV or +1.8eV for f) but never the values needed to get the reported values with the help of equation (1), and certainly not to any precision of 2 digests after the comma. have I overlooked something?
ii. Where the 4.4eV offset in equation (1) should come from needs to be explained, or a reference be given.
5. When discussing figure 3 on page 6, the 'peaks' the authors refer to in the text really seem to be the dips (local minima).
6. Figure 4a does NOT 'exhibit visible emission with peak wavelengths of 490nm and 590nm' as stated in line 183. The peaks lie somewhere else, at lower wavelengths. I also beg to differ when the authors write in lines 199-200 that 'the refractive index and extinction coefficient [...] present no considerable changes (Figure 4c and 4d)' . The red, blue and black curves are clearly distinguishable and show peak shifts (c) and resonance minima shifts (d), respectively.
7. The reader cannot understand where the energy band diagrams in figures 6b and 8d come from. These are either rough sketches or, if exact in alignment, the results of rather complicated band structure calculations not shown.
8. The statement on transmittance spectra in line 272 probably refers to Figure 8c instead of 8b.
9. The Experimental section presented does not state how much of the colloidal dispersions have been added nor how thick the resulting C-dot interlayers were. This may be critical for evaluating (or excluding) possible self-absorption of light.
Reviewer 2 Report
This research represents an important contribution to this line of work. The results may be of general interest to researchers working in the field. The article is well structured and the results obtained are convincing. However, perhaps being a permanent user of HRTEM, I would have liked the electron microscopy characterization to have been more detailed. With everything said above, I have no objections to this work, so I recommend that it be published in Nanomaterials.
Round 2
Reviewer 1 Report
The authors have adressed most issues, however, those relating to points 4 and 9 need further consideration as follows:
4a. Figure 2a lacks a calibrated scale bar. Please add to the image.
4b Figure 2b shows only one clear crystalline peak, presumably from the silicon carrier. The two indexed reflections at 17.4° and 35.6° are within noise limits and completely meaningless. This sample is probably amorphous.
4c Figure 2c indeed shows a few weak peaks. Please state what phase and lattice parameters the Miller indices refer to.
4d The Tauc plots in Figure 2d are OK but the linear extrapolations shown meaningless, and the values in the figure (2.6 and 2.3eV) do not agree with the values in the text (2.69 and 2.58eV). There are clearly several piecewise linear regions in the Tauc plots none of which would extrapolate to any of these values. Rather, the material seems to have mid-gap states in the band-gap.
4e,f While the extraction of potentials from Figures 2e&f now looks Ok, there seems to be an equation missing how to calculate LUMO energies from the reduction potential onsets measured.
9. When asked for the amount of colloidal dispersion spun onto the substrates, the data provided for rotation speed and concentration do not allow to infer how much material was actually added. Please provide either the volume or mass of liquid dropped onto a substrate area of given size. Alternatively, show a cross-section of the resulting film, which indeed must be very thin if the amount of light absorption is not changed significantly.
